# Application of Sustainable Bamboo-Based Composite Reinforcement in Structural-Concrete Beams: Design and Evaluation

**DOI:** 10.3390/ma13030696

**Published:** 2020-02-04

**Authors:** Alireza Javadian, Ian F. C. Smith, Dirk E. Hebel

**Affiliations:** 1Future Cities Laboratory, Singapore ETH-Centre, Singapore 138602, Singapore; 2Applied Computing and Mechanics Laboratory (IMAC), Civil Engineering Institute, School of Architecture, Civil and Environmental Engineering (ENAC), Swiss Federal Institute of Technology, Lausanne (EPFL), 1015 Lausanne, Switzerland; 3Department of Architecture, Karlsruhe Institute of Technology (KIT), 76131 Karlsruhe, Germany; dirk.hebel@kit.edu

**Keywords:** bamboo, fiber reinforced polymer, sustainable reinforcement, reinforced concrete

## Abstract

Reinforced concrete is the most widely used building material in history. However, alternative natural and synthetic materials are being investigated for reinforcing concrete structures, given the limited availability of steel in developing countries, the rising costs of steel as the main reinforcement material, the amount of energy required by the production of steel, and the sensitivity of steel to corrosion. This paper reports on a unique use of bamboo as a sustainable alternative to synthetic fibers for production of bamboo fiber-reinforced polymer composite as reinforcement for structural-concrete beams. The aim of this study is to evaluate the feasibility of using this novel bamboo composite reinforcement system for reinforced structural-concrete beams. The bond strength with concrete matrix, as well as durability properties, including the water absorption and alkali resistance of the bamboo composite reinforcement, are also investigated in this study. The results of this study indicate that bamboo composite reinforced concrete beams show comparable ultimate loads with regards to fiber reinforced polymer (FRP) reinforced concrete beams according to the ACI standard. Furthermore, the results demonstrate the potential of the newly developed bamboo composite material for use as a new type of element for non-deflection-critical applications of reinforced structural-concrete members. The design guidelines that are stated in ACI 440.1R-15 for fiber reinforced polymer (FRP) reinforcement bars are also compared with the experimental results that were obtained in this study. The American Concrete Institute (ACI) design guidelines are suitable for non-deflection-critical design and construction of bamboo-composite reinforced-concrete members. This study demonstrates that there is significant potential for practical implementation of the bamboo-composite reinforcement described in this paper. The results of this study can be utilized for construction of low-cost and low-rise housing units where the need for ductility is low and where secondary-element failure provides adequate warning of collapse.

## 1. Introduction

Reinforced concrete is described as the “most fruitful and generous of all building materials” by Pier Luigi Nervi, an engineer and contractor and one of the greatest 20th-century structural designers in reinforced concrete [1,2]. Over the past century, reinforced concrete has significantly transformed the built environment.

The most common type of reinforcing bar is steel reinforcement. Steel reinforcement has been used in a variety of structural concrete applications over the past decades. However, the susceptibility to steel corrosion is the major drawback. Corrosion is often initiated by either concrete carbonation or exposure of the concrete element to chloride ions, as discussed in many studies [3,4,5]. The concrete matrix creates a protection layer around the reinforcement steel member by providing an alkaline environment with a pH level of 12 to 13, where a thin oxide layer forms on the steel reinforcement prevents iron atoms from dissolving. Hence, steel reinforcement remains in a passive state and the corrosion is prevented or largely reduced [3,6,7]. The corrosion process will be initiated the moment that this protective layer around the steel reinforcement is damaged through either carbonation or exposure to chloride ions [8,9].

Various methods have been investigated by researchers worldwide to overcome the corrosion of the steel reinforcement in concrete. Improving the quality of concrete, employing protective coating on reinforcement bars, and the application of non-ferrous and non-metallic reinforcement materials are some of the many ways to reduce the problems that are associated with corrosion of steel reinforcement bars in concrete [10,11,12,13]. When comparing the methods of minimizing the impact of corrosion, technical advantages and disadvantages as well as cost need to be considered. For instance, the initial cost of applying epoxy coating on reinforcement bars on average can add up to 30% to the cost of the reinforcement, while the galvanizing process can add up to 50% to the initial cost [11,14].

Steel reinforcement is not only costly; problems that are associated with corrosion and the additional investment required to minimize corrosion have led to many challenges, particularly in developing countries. For instance, very few developing countries have the required resources of steel for development of their infrastructure. According to the report that was published by World Steel Association in 2015, out of 54 African nations, only two produce steel in noticeable quantities [15]. According to the United Nations Population Division’s report, it is expected that world population will increase to 11 billion people by 2100 and most of this increase will take place in countries in developing regions, mostly in Africa and Asia [16]. Therefore, difficult challenges will be faced by these countries to meet the demand for steel reinforcement bars for the housing and construction industry.

Steel reinforcement is not without alternatives. There is an alternative renewable natural resource that grows in the tropical zone of earth that coincides with the location of developing countries—bamboo. Bamboo belongs to the botanical family of grasses and it has relatively high tensile capacity, thus making it a potential reinforcement alternative for steel [17,18]. Some species of bamboo have shown tensile strength of up to 600 MPa and single bamboo fibers can reach a tensile strength as high as 2000 MPa [19].

The idea of employing bamboo for concrete applications is not new. In 1914, Hou-kun Chow, at MIT tested for the very first time small diameter raw bamboo as reinforcement materials for concrete members [20]. Later on, in 1935, Datta and Graf from Stuttgart investigated the potential applications of raw bamboo in concrete, However, they were not successful in implementing full-scale applications due to problems that are associated with debonding of natural bamboo from the concrete matrix as a result of water absorption and swelling [21]. More extensive research was carried out in 1950 after World War II, at Clemson University, where small diameter bamboo culms were employed as reinforcement in concrete structures [21]. However, most of the structures built with bamboo as reinforcement for concrete during this period collapsed shortly after being constructed due to shrinkage and swelling of the raw bamboo and degradation over time due to insect and fungus attacks.

Between 1995 and 2005 more recent research in the field of bamboo as reinforcement for concrete, as carried out in Brazil [22,23,24]. Seven species of bamboo were selected to find the most appropriate species to be used as reinforcement in lightweight concrete beams. The results from this study showed that concrete beams with bamboo reinforcement had significant load-bearing capacities when compared with non-reinforced beams and the strength was similar to that of concrete beams that were reinforced with steel.

Unfortunately, the long-term behavior of bamboo in concrete structures has since remained a challenge for many researchers. When natural bamboo is exposed to the concrete matrix, through time, it absorbs water from the concrete. This will result in the swelling of bamboo elements. Repetitive swelling and shrinkage of the natural bamboo has often led to sudden de-bonding of the bamboo element from the concrete matrix. This caused a near-complete loss in the structural load-bearing capacity of the reinforced concrete member.

The results from studies on the application of natural bamboo from 1914 till today have indicated that raw bamboo has potential for replacing steel in reinforced concrete beams [25,26,27,28,29,30]. However, new solutions need to be developed to overcome the problems that are associated with durability issues, such as water absorption, the effect of alkaline environment of concrete on raw bamboo, the difference in thermal-expansion coefficients between bamboo and concrete, as well as the bonding mechanism of raw bamboo with concrete matrix. Furthermore, a reinforcement system that is designed based on the physical dimensions of the raw bamboo culm will always be limited to available dimensions according to the natural characteristics of bamboo.

In an earlier study by the authors, a new technology was developed to process bamboo culms into fibers that are suitable for application in the form a novel bamboo composite material [31]. Advantages of using bamboo fibers over traditional synthetic fiber materials are the abundance, renewability, biodegradability of the fiber, and low production costs. The fibers are obtained by processing entire bamboo culms. They are then added to epoxy resin system and fabricated into high-tensile-strength composite materials while using a hot-press fabrication method. The process yields a bamboo composite material, which may then be cut into different sizes to be used in concrete as reinforcement in place of steel bars [32].

The main objective of this paper is to report on an investigation into the possibility of using this newly developed bamboo composite material for reinforcement in structural-concrete beams. The development of the bamboo composite reinforcement system and the durability properties, such as water absorption, swelling, shrinking, and chemical resistance are evaluated. Additionally, challenges that are related to thermal expansion and the bonding mechanism are studied through an extensive series of tests. Finally, comparisons of the design parameters with reference to the design guidelines stated in ACI (American Concrete Institute) 440.1R-15 for FRP (Fiber Reinforced Polymer) reinforcement in concrete [33] help to place the contributions in a more general context.

## 2. Materials and Methods

### 2.1. Bamboo Species

Dendrocalamus asper, known as Petung Putih bamboo, was selected from a bamboo forest on the Java island of Indonesia. Dendrocalamus asper is widely available in Java and is being used for small housing projects. The bamboo samples used in this study had an average culm length of 20 m. The outer diameter of the selected culms was between 70 mm and 160 mm. Figure 1 shows the type of bamboo used in this study.

In an earlier study that was carried out by the authors, the mechanical properties of Dendrocalamus asper bamboo from Indonesia with respect to culm physical properties, including culm diameter, wall thickness, height, moisture content, and specific density, were investigated and correlations between mechanical properties, including tensile strength, modulus of rupture and modulus of elasticity in flexure and tension, and culm physical properties have been studied [34]. For the purpose of this study, bamboo samples with various culm diameters and wall thicknesses were selected to ensure the average properties of the culms for composite fabrication.

### 2.2. The Matrix

There are various types of matrix that can be used for composite fabrication. The role of the matrix is to hold the dispersed fibers together and transfer the load to the fibers. The matrix phase of any fiber reinforced polymer composite can be classified as either thermoset or thermoplastic. Among the various thermosetting matrices, the epoxy system is considered to be one of the high-performance systems used mostly for advanced composites with high qualities. A commercially available two-part epoxy resin system was chosen based on three main criteria;

-Low Volatile Organic Compounds (VOCs) content;-acceptable mechanical properties;-sustainable sources.

Table 1 displays the properties of the commercially available epoxy resin system that was chosen in this study based on the criteria described earlier. 

SUPER SAP^®^ epoxy systems are produced by Entropy Resins Inc. in the US and according to the manufacturer, these contain up to 20% bio-based and renewable contents sourced from waste materials, such as wood pulp and bio-fuels production.

### 2.3. Production of Bamboo Composite Materials

The bamboo composite samples were produced following a patented processing technology that was developed by the authors [35]. Patented processing tools were developed to process bamboo culms into bamboo fiber bundles of varying thicknesses, width, and length in a detailed study that was recently carried out by the research team [31]. In this process, the bamboo sections were first boiled at 80 °C in normal water in a closed container for 8 to 20 h. The boiling helped to soften the microstructure of the bamboo sections through weakening the lignin interface’s adhesion with the cellulose fibers and, thus, the processing of the sections into fiber bundles became easier. The processed fiber bundles had thicknesses that were in the range of 0.40 mm to 1mm. The processed bamboo fibers were first dried in an air-circulated oven at 80 °C until the moisture content was less than 10%. The moisture content was measured according to the ASTM D4442-07 standard test method for direct moisture content determination of wood and wood-based materials [36]. A hand lay-up technique was used for producing the composite samples. The bamboo fibers were firstly impregnated with the epoxy resin matrix and, secondly, the layered impregnated bamboo fiber bundles were placed into the mold of a hot-press machine along the fiber direction. A simple yet effective semi-automatic hot-press compression molding machine was employed for this purpose with a maximum pressure of 25 MPa and a maximum temperature of 140 °C. Figure 2 shows the sample of bamboo composite material that was developed in this study.

### 2.4. Bamboo-Composite Reinforcement Systems

Bamboo-composite reinforcement systems for reinforcing concrete beams of up to 1300 mm in length were fabricated following the details that are explained in Section 2.3. The length of the concrete beams was chosen according to the availability of testing facilities and Universal Testing Machine’s (UTM) loading capacity of 100 kN. There were two types of reinforcement used in this study for reinforcing the concrete beams: longitudinal and transverse (shear) reinforcement. The longitudinal reinforcements were placed parallel to the long axis of the beam to provide the required tensile and flexural capacity, while transverse reinforcements were employed to provide sufficient shear strength perpendicular to the long axis of the concrete beam.

#### 2.4.1. Longitudinal Bamboo Composite Reinforcement

The longitudinal bamboo composite reinforcement elements were produced by using a steel mold that was employed within the hot press machine that had a width of 150 mm and length of at least 1200 mm. As mentioned earlier in this section, the length of the concrete beams was 1300 mm and, therefore, the additional 100 mm was designed as the cover of the reinforcements on the both ends of the reinforced concrete beams. The impregnated bamboo fiber layers were placed into the rectangular mold of hot-press and pressed at a temperature of 100 °C and a pressure of 20 MPa for 30/30 min. of curing/post-curing duration. The final composite board was cured for an additional 48 h in a curing oven at a temperature of 50 °C after removing from the hot press to ensure that the full cross-linking of the epoxy resin could take place. The final board had a length of 1,200 mm, a width of 150 mm, and a thickness of 10 mm. The final bamboo composite boards were then cut into reinforcement of size 10 × 10 × 1200 mm for reinforcing the concrete beams. The square cross section was the result of the production process of the bamboo composite materials and the rectangular steel mold of the hot press. Figure 3 shows the longitudinal bamboo composite reinforcement with a dimension of 10 × 10 × 1200 mm.

#### 2.4.2. Transverse (Stirrup) Bamboo Composite Reinforcement

The stirrups were placed perpendicular to the long axis of the beam to resist the shear forces and provide sufficient anchorage to the longitudinal reinforcements. Therefore, the load transfer within the bamboo fibers in the bamboo composite samples for stirrup production should be directed in a continuous loop, unlike in longitudinal reinforcement, in which only a unidirectional load transfer in one axis was required. To do so, a U-shape mold was fabricated from heat-treated steel for the production of the bamboo composite stirrups in this study. The cross-section of the mold was designed according to the preliminary reinforced concrete beam design before the casting and production of the concrete samples by considering the shear design concept of reinforced concrete member. Furthermore, the wide-angle corners of the cross sections of the mold gave the corners and sides of the stirrups sufficient pressure from the hot-press compression molding machine during production. This design ensured sufficient epoxy penetration into the bamboo fiber layers at the corners and along the sides of the composite samples. The stirrups were prepared by aligning the impregnated stacks of bamboo fibers along the short axis of the U-shape mold covering the two sides and bottom section of the mold. The fiber direction of the veneer layers was maintained along the sides of the steel mold, as shown in Figure 4. Similar curing/post curing conditions as used for the fabrication of the longitudinal bamboo composite reinforcement were employed.

Maintaining the fiber direction along the shorter axis of the steel mold had the advantage of achieving the highest axial strength (e.g., tensile, shear, or compressive) of the stirrups along the fiber direction by utilizing the unidirectional properties, which would give the bamboo composite stirrups their maximum mechanical properties. The final bamboo composite board of stirrup was then cut along the fiber direction into pieces with cross sections of 15 mm in width and 6mm in thickness, as shown in Figure 5. This width and thickness were chosen following the calculations of the shear capacity of stirrups with respect to the loading capacity of the UTM in the laboratory.

A closed loop shape would provide sufficient development length along both vertical (parallel) sides of the stirrups, according to the American Concrete Institute ACI 318 “Building Code Requirements for Structural Concrete and Commentary” [37]. Therefore, to prepare the closed loop stirrups, two stirrups were used and the two wide-angled single stirrups shown in Figure 5 were bent slightly, so that the two legs of the stirrups could stay parallel. The final stirrups had dimensions of 130 mm × 130 mm. The squared shape of the stirrups would lead to a square cross section for the final concrete beams. Figure 6 displays the reinforcement system that was prepared in this study consisting of longitudinal and transverse bamboo composite reinforcement. A high modulus stainless steel wire-tie system was employed to hold the stirrups together in a close loop shape, side by side, and to the longitudinal reinforcement bars. Epoxy was also used to attach the stirrups and the longitudinal bars together to prevent their movement during pouring the concrete mix.

#### 2.4.3. Properties of Bamboo Composite Reinforcement Systems

Before preparing the reinforcement bars from the final bamboo composite boards, the tensile and flexural properties, including strength and modulus of elasticity, were measured according to ASTM D3039-08, “Standard Test Method for Tensile Properties of Polymer Matrix Composite Materials” and ASTM D7264, “Standard Test Method for Flexural Properties of Polymer Matrix Composite Materials”, respectively. The compressive strength of the boards was obtained according to ASTM D6641, “Standard Test Method for Compressive Properties of Polymer Matrix Composite Materials”, along the fiber direction. For each series of tests, five samples were prepared from each composite board. A total of 10 boards were prepared for the longitudinal reinforcement and at least 50 samples were tested for each property. The samples were taken from various locations along the boards to obtain average values that represented the composite properties along the full length and width of the boards.

### 2.5. Bamboo Composite Reinforced Concrete

There are various factors that are involved in the design and proportioning of concrete beams reinforced with the newly developed bamboo composite reinforcement. The bond between bamboo composite reinforcement and the concrete matrix and the effect of water penetration as well as alkaline environment of concrete on the mechanical properties of bamboo composite reinforcement are investigated in this study.

#### 2.5.1. Evaluation of the Bond Mechanism between Bamboo Composite Reinforcement and Concrete

Acceptable bond between bamboo composite reinforcement and the concrete matrix ensures a smooth stress transfer between the two materials and, therefore, contributes to the overall load bearing capacity of the reinforced concrete element. In the event of weak bonding between reinforcement and concrete, the stress transfer between reinforcement and concrete matrix will be disrupted and, thus, failure of the member will follow, especially in the tension zone of the concrete element. Therefore, ensuring perfect and smooth bond between concrete and bamboo composite reinforcement would contribute to higher ultimate load-bearing capacity of the reinforced concrete member by activating the maximum mechanical capacities of the bamboo reinforcement through providing an interfacial microstructure that would ensure continuous tensile stress transfer between the two materials.

A detailed study was previously carried out by the authors where a series of pull-out tests were designed to find the most suitable method to enhance the bonding between the two materials to understand the bonding mechanism between bamboo composite reinforcement and concrete, [32]. In the work that was carried out by the authors, four types of coatings were investigated by applying them on the surface of the reinforcements before being placed into the concrete cylinders to evaluate the bond strength through pull-out tests. The work that was carried out by [32] showed that an epoxy-based coating with the addition of sand particles provided extra protection without loss of bond strength.

Therefore, a two-component, water-based epoxy system was chosen in combination with sand particles. The waterproof system would block the moisture and water from the interface of the concrete and bamboo composite reinforcement, thus the stress transfer between concrete and reinforcement would be enhanced. Furthermore, the addition of sand particles could improve the surface roughness of reinforcement, which results in an improved bonding strength with the concrete matrix through creating physical and mechanical interlocking systems with the aggregates of the existing concrete matrix [32].

Normal strength concrete with an average compressive strength of 20 MPa was used to prepare the concrete samples for the beam and pull-out tests. Half dog-bone shaped bars with a cross section size of 10 mm × 10 mm were prepared from the bamboo composite boards that were produced in this study. The dog-bone shape for the grip of the UTM machine helped to eliminate the risk of slippage during the tests. An embedment length of 200 mm was used in this study, which showed the highest bending strength in the earlier work that was carried out by the authors [32]. Cylinders of 300 mm in height and 150 mm in diameter were used to prepare the concrete pull-out specimens. The recommendations of ASTM C900-15 standard titled “Standard Test Method for Pull-out Strength of Hardened Concrete” were largely followed. Figure 7 shows the pull-out sample and Table 2 shows the test set-up and the concrete mix for a volume of 1 m^3^ concrete.

Average tensile strength and elastic modulus in compression of the concrete samples were 6.8 MPa and 26,100 Mpa, respectively, after 28 days of curing. For the purpose of the study’s pull out tests have been carried out by preparing a custom-made insert for the Shimadzu UTM machine of type AG-IC 100 kN following the works that were described by [32]. The setup was made from hardened steel to hold the position of the concrete cylinder during the pull-put test. The loading rate was set to 2 mm/min. Figure 8 shows a schematic representation of the forces that are associated with the bond strength between the concrete and the bamboo composite reinforcement. P is the pull-out force, l_a_ is the embedment length in mm, τ is the bond strength in MPa, and *a* and *b* are the cross-sectional dimensions of the reinforcement in mm.

Equilibrium in Figure 8 leads to the following:(1)$$\tau .(2a+2b).l_{a}=P$$

From Equation (1), the bond strength, τ, is determined while using Equation (2):(2)$$\tau =\frac{P}{(2a+2b).l_{a}}$$

A total of 20 pull-out specimens were tested in this study.

#### 2.5.2. Durability Assessment of the Bamboo Composite Reinforcement

The alkaline environment of concrete could potentially affect the properties of bamboo composite reinforcement when used in concrete. There are several factors that might have negative impacts on the physical and mechanical properties of bamboo composite reinforcement in concrete. Water penetration occurs over time through the concrete matrix when it is exposed to humid air and rain. In this study, the effect of water penetration and alkaline environment of concrete on the mechanical properties of bamboo composite reinforcement is investigated through a series of laboratory tests.

##### Water Absorption Properties

The samples of 10 × 10 × 10mm in size were prepared from bamboo composite boards for water absorption tests. The samples were immersed in distilled water with a temperature of 23 °C and 60 °C. The higher temperature was chosen to evaluate the effect of water uptake at higher temperature on the properties of bamboo composite reinforcement in comparison with room temperature water absorption. The water absorption tests were carried out in accordance with ASTM D5229M-14 “Standard Test Method for Moisture Absorption Properties and Equilibrium Conditioning of Polymer Matrix Composite Materials”. The weight changes were recorded every 2 h during the first week of the experiment when the water uptake rate was relatively high; subsequently, measurements were only carried out once per day for up to three months’ immersion in water (2160 h) when the rate of water absorption was reduced. 10 samples were prepared for each testing condition. At various stages of soaking, the percentage of the weight gain was calculated by measuring the difference in weight between the dry condition and the weight after water immersion

##### Alkali Resistance Properties

The alkaline resistance of bamboo composite reinforcement in concrete was investigated according to ASTM D7705-12 “Standard Test Method for Alkali Resistance of Fiber Reinforced Polymer (FRP) Matrix Composite Bars used in Concrete Construction”. The tensile test samples were prepared according to ASTM D3039-08 and coated with the water-based epoxy coating before being subjected to an alkaline solution consisting of 118.5 g of Ca(OH)_2_, 0.90 g of NaOH, and 4.2 g of KOH in one liter of tap water with an initial pH value of 12.6 to 13.0, similar to the pH value of the pore water inside the concrete matrix. The samples were immersed in the alkaline solution at 60 °C for a period of up to three months (2160 h). At different intervals, the samples were removed from the alkaline solution and carefully washed with tap water to remove the excess alkaline solution before measuring their tensile strength and modulus of elasticity in tension. The tensile capacity retention was calculated according to Equation (3).
(3)$$R_{et}(\%)=\frac{F_{tu1}}{F_{tu0}}\times 100$$
where:
*R_et_* = tensile capacity retention (%)*F_tu0_* = initial tensile capacity before immersion in alkaline solution, in Newton*F_tu1_* = tensile capacity after immersion in alkaline solution, in Newton

#### 2.5.3. Design of the Bamboo Composite Reinforced Concrete Beams according to ACI 440.1R-15

The Bamboo composite reinforcement system has different behavior when compared with steel reinforcement in terms of the physical and mechanical properties. The behavior of bamboo composite reinforcement systems is similar to typical FRP reinforcement that is produced with carbon fiber (CFRP) or glass fiber (GFRP). Bamboo composite materials display an elastic behavior up to the ultimate failure point and do not yield like steel.

In general, bamboo composite reinforcement displays low stiffness when compared with traditional construction materials. Therefore, it is difficult to compete in the mass construction market, where, for instance, the deflections of the building can be critical for medium to high rise structures. Such structures would likely develop cracks due to the lower elastic modulus of the bamboo composite reinforcement. However, there are applications where deflections are not critical in the design, such as low rise, low cost housing solutions for developing countries where the demand for ductility is low and the failure of secondary elements will provide adequate warning of collapse.

ACI 440.1R-15 “Guide for the design and construction of structural concrete reinforced with Fiber Reinforced Polymer (FRP) bars” has provided design guides for the application of FRP materials as reinforcement in concrete. These guides account for the lower ductility of the FRP reinforced concrete when compared with steel reinforced concrete members.

Given the consideration of ductility by ACI 440.1R-15, this design guide was the primary and most relevant guide to be used in this study for evaluating the performance of bamboo composite reinforced concrete beams. The design guides provided by ACI 440.1R-15 are based on the concepts of equilibrium and compatibility (American Concrete Institute, 2015). The ultimate strength design method was preferred over the working stress design approach to achieve comparable results with the methods employed by other standards, such as ACI 318 for steel reinforced concrete design, to design the concrete beams reinforced with bamboo composite reinforcement.

The serviceability limit state could govern the design of concrete members reinforced with bamboo composite materials to prevent excessive deflections as a result of lower ductility given the lower modulus of elasticity of bamboo composite reinforcement compared to steel. Furthermore, ACI 440.1R-15 recommends applying reduction factors to the guaranteed tensile strength to include the effect of various environmental conditions on the properties of FRP reinforcement.

In this study, a conservative reduction factor of 0.80, similar to that for GFRP reinforcement not exposed to earth or extreme weather conditions, has been used for designing the concrete beams for laboratory conditions. No reduction factor has been recommended by ACI 440.1R-15 for the modulus of elasticity, thus a similar reduction factor (0.80) has been used for the design modulus of elasticity of the bamboo composite reinforcement. A similar reduction factor (0.80) was also employed for the rupture strain to calculate the design rupture strain for the determination of the serviceability limit states of design. The average guaranteed rupture strain measured during the tensile tests was 1.35% for bamboo composite reinforcement.

In the case of bamboo composite reinforced concrete member, no ductile behavior is observed when compared to steel reinforcement. Significant elastic elongation is expected for the concrete beams that were reinforced with bamboo composite reinforcements. According to ACI 440.1R-15 and depending on the reinforcement ratio, two modes of failure are expected for bamboo composite reinforced concrete members: bamboo composite reinforcement rupture and concrete crushing according. If the reinforcement ratio is less than the balanced reinforcement ratio, the bamboo composite reinforcement rupture occurs first; otherwise, concrete crushing controls the failure mode.

ACI 440.1R-15 has recommended using the following formulae to calculate the design tensile strength of FRP bars at a bend and in this study for the bamboo composite stirrups.
(4)$$f_{fb}=(0.05\times \frac{r_{b}}{d_{b}}+0.30)f_{fu}\leq f_{fu}$$
where:
*f_fb_*: is the design tensile strength of the bend portion of FRP reinforcement (MPa),*r_b_*: is the radius of the bend (mm),*d_b_*: is the diameter of reinforcing bar (mm), and*f_fu_*: is the design tensile strength of FRP reinforcing bar after employing the reduction factor (MPa).

The nominal flexural strength of FRP or bamboo composite reinforced concrete member can be calculated while using the following formula that was recommended by ACI 440.1R-15 only when the concrete crushing limit state controls the failure.
(5)$$M_{n}=\rho_{f}f_{f}\left[1-0.59\frac{\rho_{f}f_{f}}{f_{c}{}^{’}}\right]bd^{2}$$
where *f_f_* is the stress in reinforcement in tension in MPa and can be computed according to the recommendation of ACI 440.1R-15.

In the case of bamboo composite reinforcement rupture mode of failure, the nominal flexural strength can be determined by using the following formulae as per the recommendation of ACI 440.1R-15.
(6)$$M_{n}=A_{f}f_{fu}(d-\frac{\beta_{1}c}{2})$$

This formula involves the unknown parameter (c), which is the depth to the neutral axis of the concrete cross section. A minimum amount of reinforcement is required for the section during the design process to ensure adequate resistance is provided to the minimum tensile stresses developed as a result of external loadings to prevent sudden failure of the reinforced member due to concrete crushing. The minimum amount of reinforcement for FRP reinforced concrete members according can be found according to ACI 440.1R-15.

Besides cracking, the serviceability limit state needs to be satisfied according to ACI 440.1R-15, where for a simply supported beam one-tenth of the span is recommended as the minimum thickness. Therefore, in this study for a simply supported reinforced concrete beam with a span length of 1050 mm, the minimum required depth is 105 mm. Furthermore, ACI 440.1R-15 recommends the determination of the cracking moment (*M_cr_*) and evaluation of the flexural moment when the member has cracked after deflection, given the lower modulus of elasticity of FRP. ACI 440.1R-15 recommends using the cracked moment of inertia ($I_{cr}$) to calculate the cracking moment.

Given the variation in stiffness along the length of the reinforced concrete member, ACI 440.1R-15 recommends using an effective moment of inertia (*I_e_*) to estimate the deflection of FRP reinforced concrete beams, according to the following formulae.
(7)$$I_{e}=\frac{I_{cr}}{1-\gamma {\left(\frac{M_{cr}}{M_{a}}\right)}^{2}\left[1-\frac{I_{cr}}{I_{g}}\right]}\leq I_{g}\;\mathrm{Where}\;M_{a}\geq M_{cr}$$
where: *M_a_* is the maximum service load moment and *ϒ* accounts for the variation in stiffness along the length of the concrete member and it can be calculated by the following formulae.
(8)$$\gamma =1.72-0.72(\frac{M_{cr}}{M_{a}})$$

Further calculation of cracking moment is described in ACI 440.1R-15. Through the procedure described in ACI 440.1R-15, the initial size of the cross section of the beam is estimated and the deflection limits are checked. However, the shear design of the bamboo composite reinforced concrete beam should also be controlled by providing sufficient stirrups at regular spacing. Concrete shear resistance of FRP reinforced concrete members can be computed according to ACI 440.1R-15. 

Furthermore, ACI 440.1R-15 recommends using the following to check the tensile strength of the FRP stirrups for shear design.
(9)$$f_{fv}=0.004E_{f}\leq f_{fb}$$
where *f_fb_* is the tensile strength of the bent portion of FRP stirrup, which can be calculated according to the following formula.
(10)$$f_{fb}=(0.05\times \frac{r_{b}}{d_{b}}+0.30)f_{fu}\leq f_{fu}$$
where $\frac{r_{b}}{d_{b}}$ is the ratio of the internal radius of the bent portion of the stirrup to the diameter of the FRP stirrup. Therefore, the required spacing (*s*) between the stirrups along the reinforced concrete beam can be computed by following formulae.
(11)$$\frac{A_{fv}}{s}=\frac{\left(V_{u}-\phi V_{c}\right)}{\phi f_{fv}d}$$

The design and calculations of the concrete beams reinforced with bamboo composite reinforcement are carried out according to ACI 440.1R-15, as it is the only relevant design standard available for fiber reinforced composite materials used as reinforcement in concrete, as stated earlier for the purpose of this study.

#### 2.5.4. Bamboo-Composite Reinforced-Concrete-Beam Preparation and Testing

In this study, the cross section of the concrete beams was designed according to the dimensions of the produced bamboo composite reinforcement system and the limitation of the UTM and its capacity, as explained earlier. Closed-loop testing under deformation control was performed on bamboo composite beam having dimensions of 130 × 130 mm. A concrete cover of 15mm around the stirrups following the recommendations of ACI 440.1R-15 was chosen, which results in a concrete cross section of 160 × 160 mm, as shown in Figure 9.

A total of 16 concrete beams were prepared based on three parameters affecting their design, including the number of longitudinal bars at the bottom of the beam, spacing of the stirrups, and the distance between the load introduction points in the middle section of the beam. The four-point loading set-up was used in this study to allow for a zero shear zone along the middle section of the bamboo composite reinforced concrete beam. Figure 10 shows a typical shear (V) and moment (M) distribution of the applied loads (P) during the four-point flexural test. The zero shear zone permits the elimination of the shear reinforcement, thus the longitudinal reinforcement is completely loaded in tension and flexure.

Table 3 summarizes the arrangement of the reinforcement and the loading distance according to the respective beam design. Four samples were prepared and tested per design. Two reinforcement bars were used as top compression reinforcement for all of the beams. During each test, ultimate failure load, ultimate flexural capacity (MOR), load corresponding to first crack, and flexural capacity at the time of first crack were obtained. Figure 11, Figure 12, Figure 13 and Figure 14 display the sketches of the beam designs that were prepared in this study. All of the dimensions are in mm.

For comparison purposes, four concrete beams without reinforcement were also tested in this study.

## 3. Results

### 3.1. Properties of Bamboo Composite Reinforcement

Tensile and flexural properties, including strength and modulus of elasticity of the bamboo composite reinforcement, were measured according to ASTM D3039-08, while the compressive strength of the samples was obtained according to ASTM D6641. The average properties of the Dendrocalamus asper, known as *Petung Putih* bamboo, used in this study have been previously reported by the authors, as shown in Table 4. The variation in the properties are the result of the tests carried out on different sections of the bamboo culm, as explained by the authors in their previous work [34].

Table 5 displays the average mechanical properties and standard deviations of the final bamboo composite reinforcement systems. The average specific density of both longitudinal and transverse reinforcement systems was 1.33.

The average mechanical properties of bamboo composite reinforcement materials are higher than the average mechanical properties of raw bamboo, except in the case of transverse reinforcement, where the tensile strength is slightly lower than the upper limit of the strength of the raw bamboo, as shown in Table 5. All of the mechanical properties that were measured for the transverse reinforcement were lower in comparison with longitudinal-reinforcement mechanical properties. The reduction of mechanical properties for transverse reinforcement was due to the production process of the stirrups in comparison with the production of longitudinal reinforcement. The inclined side legs of the stirrups received less pressure when compared to the bottom flat sections of the steel mold due to the wide angle chosen for the mold on the two sides, given the complexity of the shape required for the steel mold during the hot-press compression molding process. The reduced pressure on the inclined sides of the steel mold would subsequently lead to lesser penetration of the epoxy matrix into the bamboo fiber layers. Thus, the interface between the epoxy and fibers could not be completely developed to its full strength compared to longitudinal reinforcement, resulting in lower mechanical properties.

As explained in earlier with regards to the fabrication of the stirrups, the inclined side legs of the stirrups received less pressure when compared with the bottom flat sections laid on the bottom side of the steel mold due to the wide angle chosen for the mold on the two sides due to the complexity of the shape required for the steel mold during the hot-press compression molding process. The reduced pressure on the inclined sides of the steel mold subsequently led to lower penetration and infusion of the epoxy matrix into the veneer layers. Thus, the interface between the epoxy and fibers was not completely developed to its full strength due to the lack of complete infusion of the epoxy matrix into the various layers of bamboo fibers. As a result, the weak interfacial bond strength between bamboo fibers and epoxy system subsequently led to lower mechanical properties. However, the results in Table 5 show that the novel techniques for processing the bamboo into fiber bundles together with new production methods used in this study improved the mechanical properties of the final bamboo composite as compared to raw bamboo sections. This has also been observed by [31,34,38,39]. For instance, when the modulus of rupture of the longitudinal bamboo composite reinforcement is compared with the upper limit of the mean modulus of rupture values for raw bamboo, an improvement of up to 27% is observed. Similarly, the tensile strength and modulus of elasticity in tension of the longitudinal bamboo composite reinforcement are enhanced when compared with the upper limit of the mean values for raw bamboo by up to 7% and 26%, respectively.

### 3.2. The Bond Strength between Bamboo Composite Reinforcement and Concrete

In this study a two-component, water-based epoxy system was chosen in combination with sand particles to enhance the bonding mechanism between bamboo composite reinforcement and concrete, while an embedment length of 200mm was used following the earlier work that was carried out by the authors, as discussed in Section 2.5.1. The maximum failure load was recorded in each test and the bond strength was determined while using Equation (2). Table 6 displays the average and standard deviation of the bonding strength together with the observed mode of failure.

The results show that similar bond strength values were obtained through the techniques used in this study, as reported by [32]. The bond strength and the mode of failure were in accordance with the results of the earlier studies. The embedment length of 200mm was also sufficient for developing the necessary bonding strength between bamboo composite reinforcement and concrete. Therefore, to reinforce the concrete specimens in this study with bamboo composite reinforcement, the same technique was employed to ensure sufficient bond strength between the two materials.

### 3.3. Water Absorption Properties

Water absorption tests were carried out in accordance with ASTM D5229M-14 while the weight changes were recorded every 2 h during the first week of the experiment and, subsequently, measurements were only carried out once per day for up to three months’ immersion in water (2160 h). Figure 15 displays the average water absorption behaviour in percentage by mass for the three-month measurement for the two temperature conditions.

The water absorption of the bamboo composite reinforcement samples reached a maximum of 0.5% of the weight of the dry sample. The low rate of water absorption indicates the high resistance of the bamboo composite samples to water and moisture ingress, even at extreme conditions. A quasi-equilibrium state was achieved in both 23 °C and 60 °C temperatures after 170 h of immersion in water. However, the water absorption values for samples that were immersed in water at a temperature of 60 °C were slightly higher than samples that were immersed in room temperature water for a similar period of time. This difference could be attributed to the moisture diffusion behaviour into the composites, in which the polymer matrices are affected by the temperature. At higher temperatures, the diffusion rate tends to be influenced by the faster movement of the individual molecules when compared to lower temperatures; therefore, the moisture or water absorption would increase as a result of a higher diffusion rate.

Nevertheless, water-based coating was applied on the surface of all reinforcements when used in concrete to minimise humidity infusion and water absorption effects on the integrity of the bamboo composite reinforcement. The water-based coating had two functions; firstly, enhancing the bonding strength of the bamboo composite reinforcement and, secondly, deterring the water from infusing into the bamboo composite reinforcement while in concrete.

### 3.4. Alkali Resistance Properties

The alkaline resistance of bamboo composite reinforcement in concrete was investigated according to ASTM D7705-12. The tensile capacity retention was calculated according to Equation (3). Figure 16 displays the tensile capacity retention (*R_et_*) for the tensile strength and modulus of elasticity of the samples. The reduction in both tensile properties followed a similar trend. The modulus of elasticity was shown to be more influenced by the exposure to alkaline solution over time. However, it has been observed that tensile properties reach a steady state after one month’s exposure to the alkaline solution.

### 3.5. Properties of the Bamboo Composite Reinforced Concrete Beams

A total of 16 concrete beams were prepared based on three parameters affecting their design, including the number of longitudinal bars at the bottom of the beam, the spacing of the stirrups, and the distance between the load introduction points in the middle section of the beam. Table 7 displays the results of the flexural tests for the 20 concrete beams.

The initial cracking load where the first crack was observed in the concrete beam started to increase by the increase in the number of bottom reinforcement while maintaining the same load introduction points. The lowest initial cracking load was observed for non-reinforced concrete beams and the largest initial cracking load was observed with beam design C samples with four reinforcement bars at the tension face of the concrete cross section and four stirrups on each side of the beams. It was observed that, by reducing the spacing between the stirrups, initial cracking and ultimate failure load were both improved. This is the result of the enhancement in shear span of the beam by preventing the shear failure and, thus, the bamboo composite longitudinal reinforcement could be activated up to its ultimate tensile capacity.

In general, three modes of failure were observed: rupture of bamboo composite longitudinal reinforcement in tension, crushing of concrete in compression, and shear failure of the beams. In concrete beam series of “A” with a stirrup spacing of 70 mm and two bottom tension reinforcement, two concrete beams experienced shear failure, while two concrete beams had a concrete crushing mode of failure. By increasing the number of bottom longitudinal bars with similar spacing of stirrups or with an increase in spacing while maintaining the four bars at the bottom side of the beam the shear capacity was improved to the point that no shear failure was observed within the samples of concrete beams belonging to the “B”, “C”, and “D” series.

The concrete crushing mode of failure was caused by the high amount of compressive stresses at the distance between the load introduction points. The concrete beams belonging to the “B”, “C”, and “D” series experienced crushing mode of failure in concrete. Furthermore, the concrete crushing was accompanied by the tensile rupture of the bamboo composite reinforcement at the tension face of the beam. The various spacing of stirrups that were used in this study and the failure modes observed showed the importance of stirrups in preventing the shear failure and the effectiveness of newly developed bamboo composite stirrups in enhancing the shear capacity of the concrete beams when used with appropriate spacing.

Figure 17a,b, respectively, show the load-displacement curves for bamboo composite reinforced concrete beam and non-reinforced concrete beam. The failure modes observed in this study are brittle failure modes in comparison with steel reinforced concrete beam in which the yielding behavior of steel reinforcement allows for a ductile behavior of concrete beam. The ACI 440.1R approach in design is to prevent the longitudinal bamboo composite reinforcement rupture or shear failure of the stirrups by having sufficient amount of tensile reinforcement in order to have concrete crushing mode of failure at the first place. This method of design ensures that there will be no sudden failure of concrete beam, but there will be some warning signs from concrete crushing before the final failure when the rupture of the bamboo composite longitudinal reinforcement takes place.

Furthermore, when the maximum mid-span deflection of the beams was measured during the tests, it was observed that the mid-span deflection decreased by increasing the amount of tensile reinforcement Figure 18 displays the average maximum mid-span deflection of the beams tested. The non-reinforced concrete beams’ maximum mid-span deflection was found to be lower than the bamboo composite reinforced concrete beams. This is the result of the lower ultimate failure load that was experienced while testing the non-reinforced concrete beams and it was not a result of the stiffness characteristics of the beam. The non-reinforced beams failed much earlier than bamboo composite reinforced concrete beams and at lower ultimate failure loads, thus smaller deflection was recorded during the tests. In general, it was observed that bamboo composite reinforcement was efficient in enhancing the ultimate load capacity of the concrete beams, while also improving the stiffness of the beams by the addition of more reinforcement bars at the tension side of the beams.

A series of calculations were also carried out to estimate the failure load and deflection limits to compare the results obtained from the tests on the ultimate failure load and the maximum deflection as well as the mode of failure with the calculations specified in ACI 440.1R-15.

## 4. Discussion

Concrete beams from design “A” Series display the lowest failure load that was obtained both from the experiments and ACI design guide. However, the cracking loads obtained from the ACI design guide remain the same for beams from design “A”, “B”, and “C” series, while beams from design “D” series display the lowest ACI cracking load. This indicates the effect of reinforcement ratio on the ultimate failure load and the cracking load. Furthermore, the interaction of longitudinal reinforcement and stirrups helps to enhance the shear capacity of the bamboo composite reinforced concrete beams. The shear capacity of the beams plays an important role in determining the ultimate failure modes. The ACI 440.1R-15 standard enforces the required spacing of the stirrups to prevent any potential shear failure of the beam and to activate either concrete crushing or reinforcement rupture as the desirable modes of failure. Only beams from design “A” series have experienced shear failure mode while no shear failure was observed within the other design series, as described earlier.

Figure 19 displays the comparison of the failure and cracking load results that were obtained in this study from the experiments and ACI standard. As shown in Figure 19, bamboo composite reinforced concrete displays better initial cracking load and higher ultimate load-bearing capacity in comparison with the design values obtained through calculations according to ACI 440.1R-15. The estimated design cracking loads that were obtained according to ACI 440.1R-15 are lower than the experimental values obtained by testing the bamboo composite reinforced concrete beams. The cracking loads that were measured during the flexural test of the bamboo composite reinforced concrete beams, on average, are larger than the design values of the ACI 440.1R-15 standard, thus confirming the superior performance of the bamboo composite reinforced concrete in comparison with the estimates according to the ACI standard. Furthermore, the results indicate the higher reserve capacity of strength that the bamboo composite reinforcement has to offer in comparison with FRP reinforcement according to the ACI design guide. For instance, when the distance between the load introductions points are increased from 350mm to 600mm, the reserve flexural capacity of the bamboo composite reinforcement can be activated, developing a balance stress transfer from the concrete matrix to the bamboo composite reinforcement within the beam. This reserve strength is largely effective in maintaining the overall load-bearing capacity of the bamboo composite reinforced concrete beams in comparison with the FRP reinforced concrete beams that are designed according to the ACI guidelines.

This finding points to the importance of the safety factors and strength reduction factors in the design and evaluation of bamboo composite reinforced concrete beams, which are in the scope of the ACI standard. The safety factors and strength reduction factors allow the engineer to include uncertainty in materials properties, including tensile and flexural strength. In the case of bamboo composite reinforcement, where a natural-based fiber (bamboo) composite material was employed as the main load-bearing element, the safety factors and strength reduction factors should also be incorporated in the design, similar to the recommendation that is given in ACI 440.1R-15 for FRP reinforcement.

The deflection results, as shown in Figure 20, also confirms the effect of lower elastic modulus of bamboo composite reinforcement in comparison with steel reinforcement and some of the FRP reinforcement on the deflection behavior. Lower stiffness of bamboo composite reinforcement is the main reason for the relatively higher deflection values that were obtained during the flexural test. However, it is shown that by increasing the reinforcement ratio from two tensile reinforcement elements to four tensile reinforcement elements, a significant increase in the overall stiffness of the beams is possible.

## 5. Conclusions

Earlier studies on the application of natural bamboo from 1914 have indicated that raw bamboo has the potential for replacing steel in reinforced concrete beams, However, problems that are associated with durability have impeded the widespread use of bamboo in the construction industry. This research demonstrates the potential of the newly developed bamboo composite material for use as a new type of element for non-deflection-critical applications of reinforced structural-concrete members. Durability is greatly enhanced since fibers are embedded in epoxy. The results of this study lead to the following additional conclusions:Flexural test results demonstrate the suitability of the ACI 440.1R-15 guidelines for the application of newly developed bamboo composite reinforcement in structural concrete beams.The results of the concrete beam test series carried out in this study validate the expected performance that has been indicated from small-specimen tests of tensile and pull-out properties of bamboo composite materials.The results of this study can be utilized for the design of low-cost and low-rise housing units where reinforcing steel is hard to obtain, where the demand for ductility is low, and where secondary-element failure provides adequate warning of collapse.

## Figures and Tables

**Figure 1 materials-13-00696-f001:**
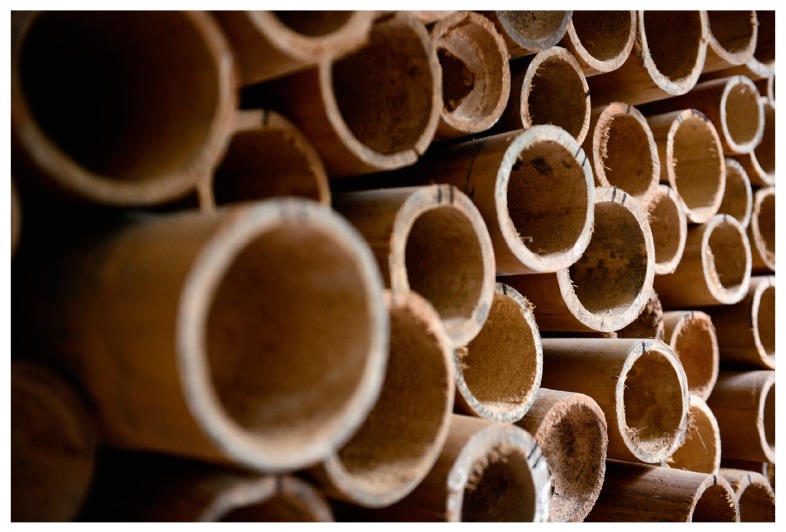
Bamboo Dendrocalamus asper used in this study to fabricate the bamboo composite materials.

**Figure 2 materials-13-00696-f002:**
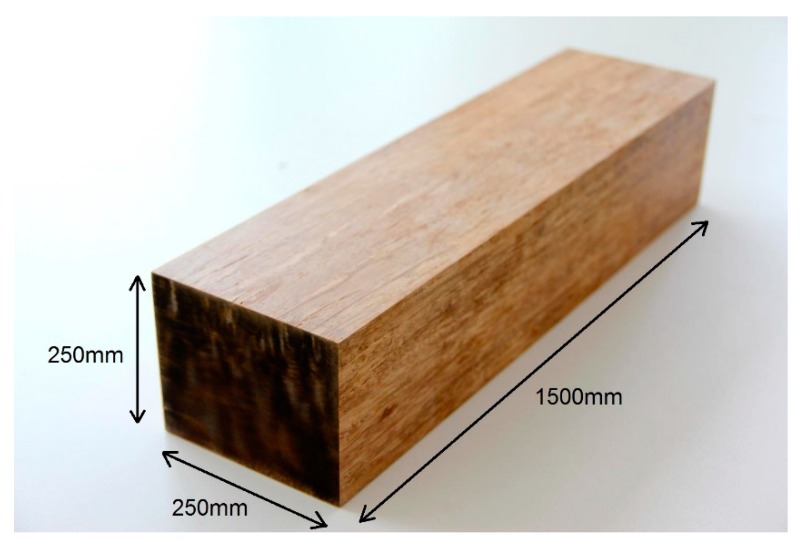
Bamboo composite sample developed in this study.

**Figure 3 materials-13-00696-f003:**
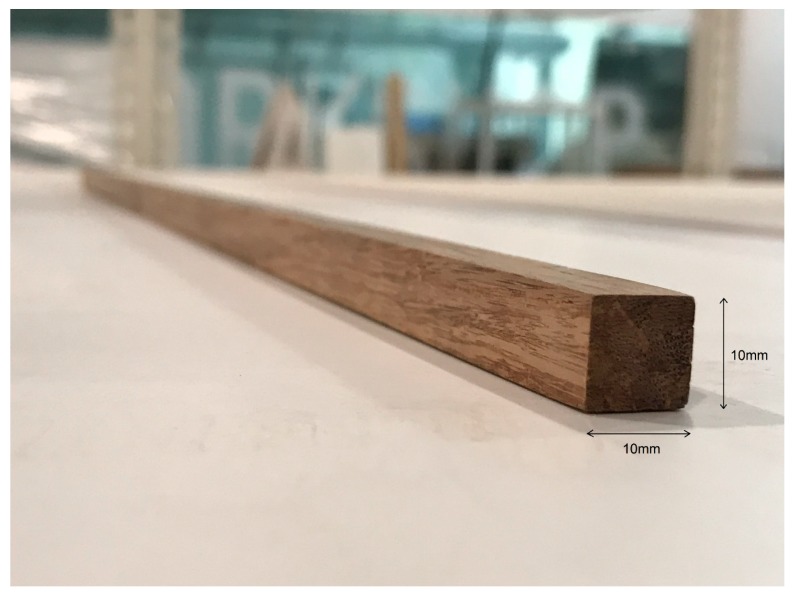
Longitudinal reinforcement made from bamboo composite material.

**Figure 4 materials-13-00696-f004:**
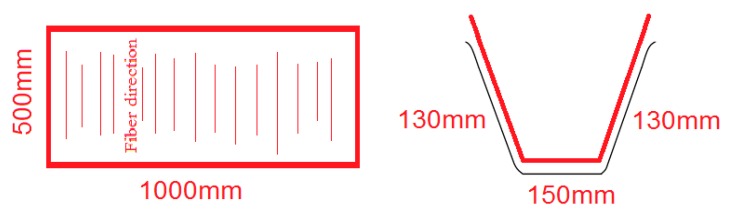
A schematic of bamboo-fiber placement into U-shaped molds for stirrup manufacture.

**Figure 5 materials-13-00696-f005:**
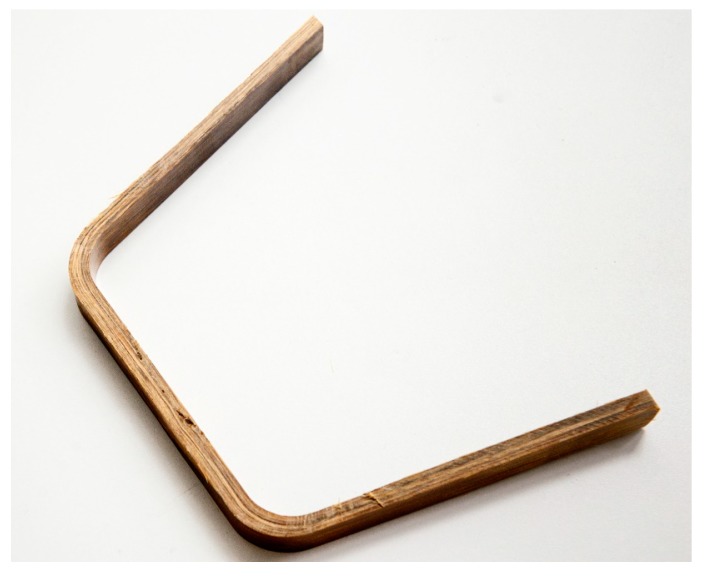
Transverse reinforcement (stirrup) made from bamboo composite material.

**Figure 6 materials-13-00696-f006:**
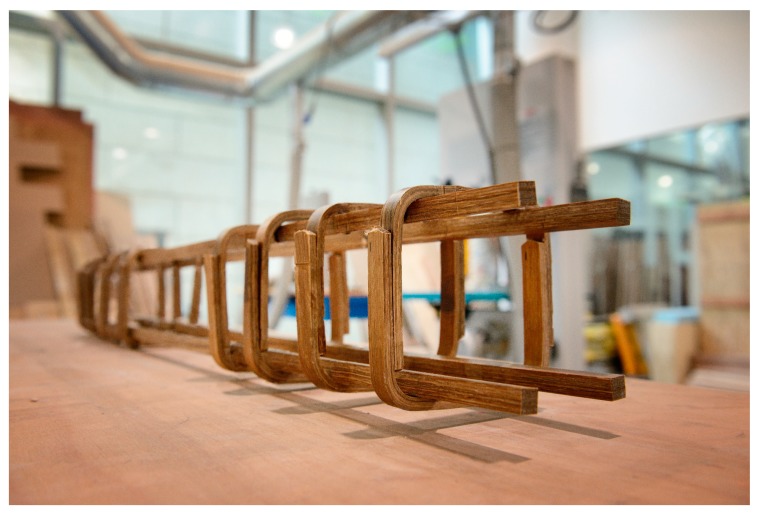
Bamboo composite reinforcement system.

**Figure 7 materials-13-00696-f007:**
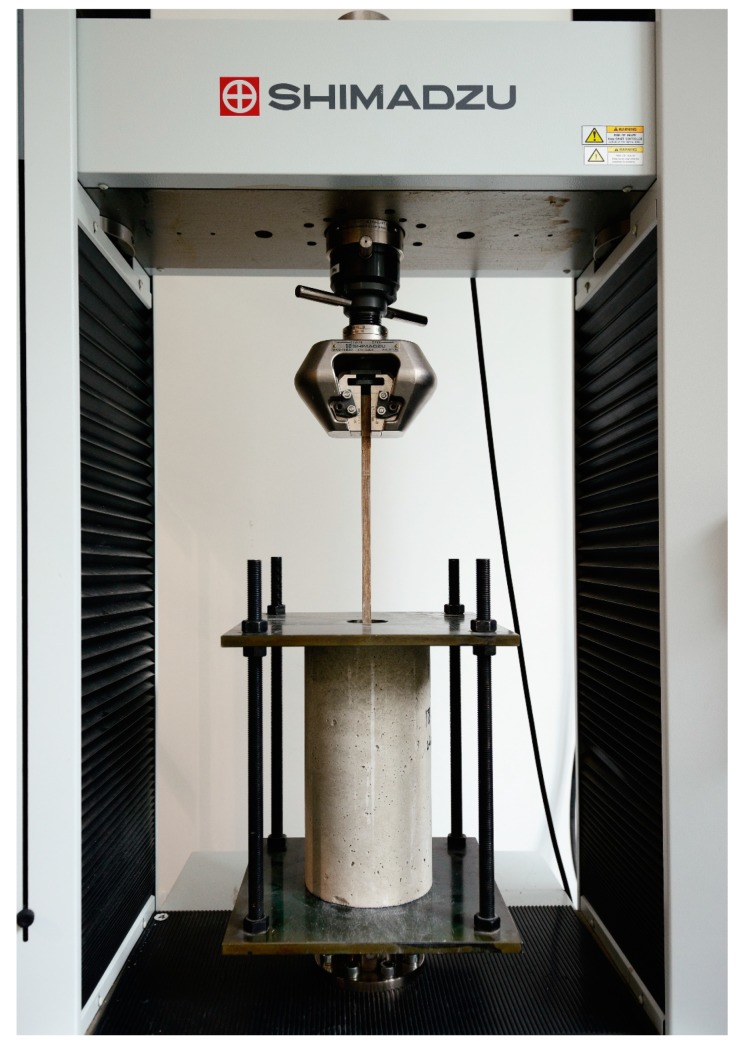
Pull-out test setup to evaluate the bond strength of bamboo composite reinforcement to concrete.

**Figure 8 materials-13-00696-f008:**
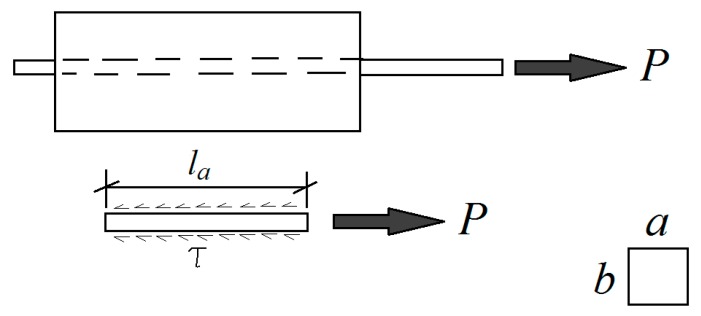
Equilibrium of forces in bond strength evaluation of bamboo composite bar and concrete during the pull-out test [32].

**Figure 9 materials-13-00696-f009:**
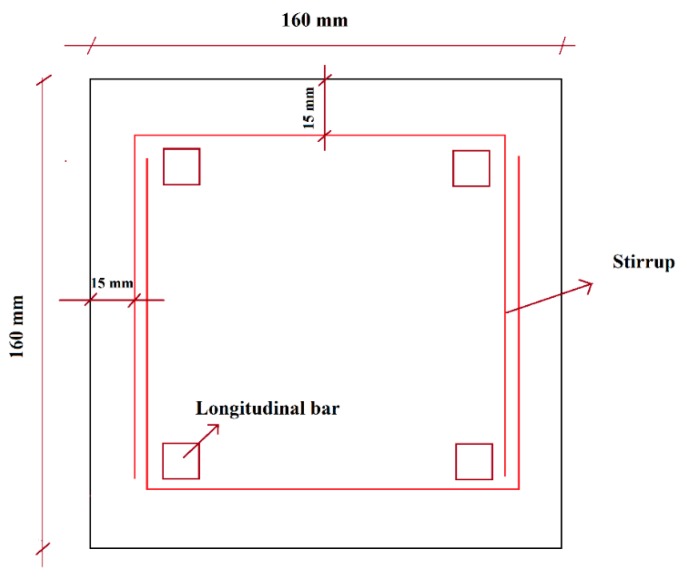
Cross section of bamboo composite reinforced concrete beam.

**Figure 10 materials-13-00696-f010:**
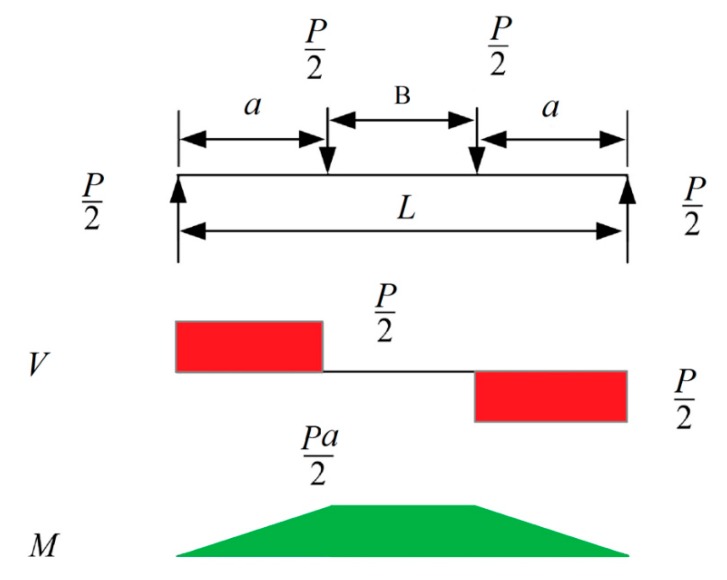
Typical load distribution for a four-point loading set-up.

**Figure 11 materials-13-00696-f011:**
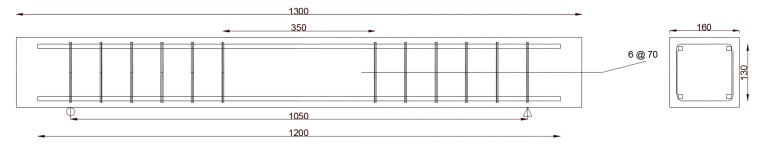
Design of beam A.

**Figure 12 materials-13-00696-f012:**
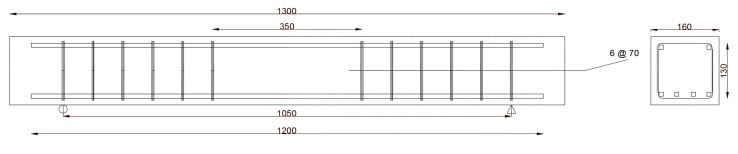
Design of beam B.

**Figure 13 materials-13-00696-f013:**
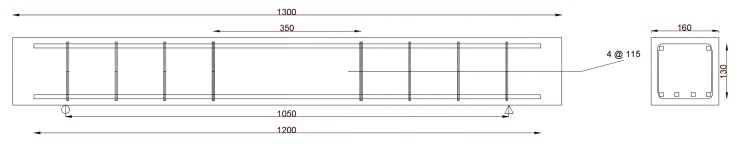
Design of beam C.

**Figure 14 materials-13-00696-f014:**
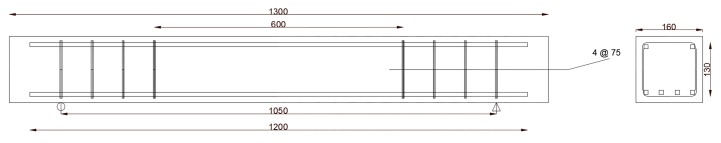
Design of beam D.

**Figure 15 materials-13-00696-f015:**
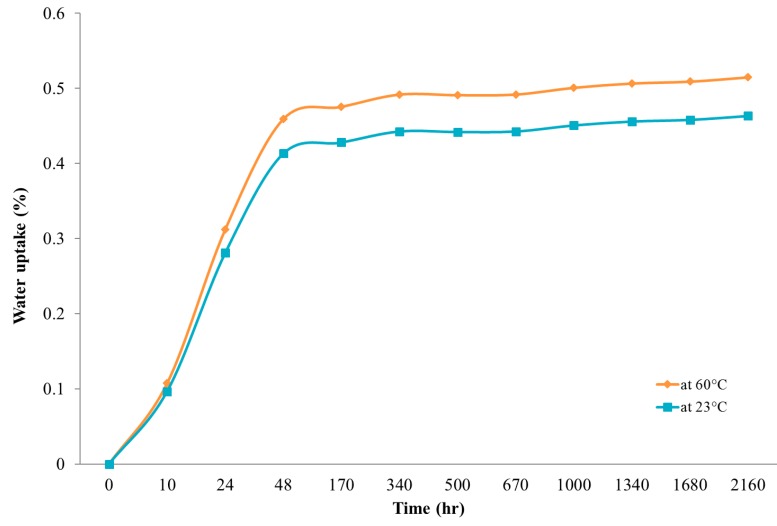
Measurement of water absorption of bamboo composite reinforcement for up to three months.

**Figure 16 materials-13-00696-f016:**
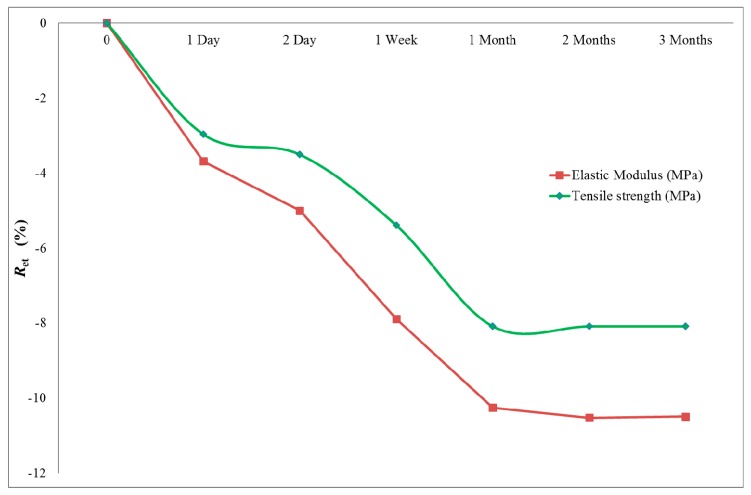
Tensile capacity retention (R_et_).

**Figure 17 materials-13-00696-f017:**
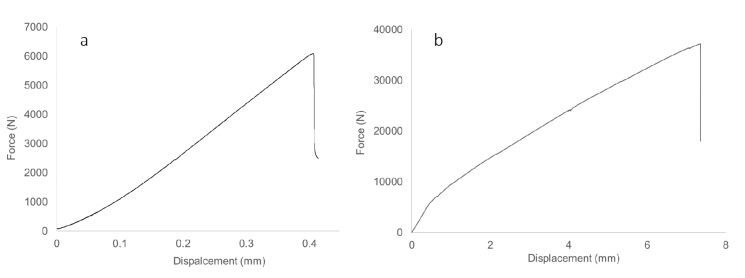
Typical load displacement curves for (**a**) non-reinforced and (**b**) bamboo composite reinforced concrete beam.

**Figure 18 materials-13-00696-f018:**
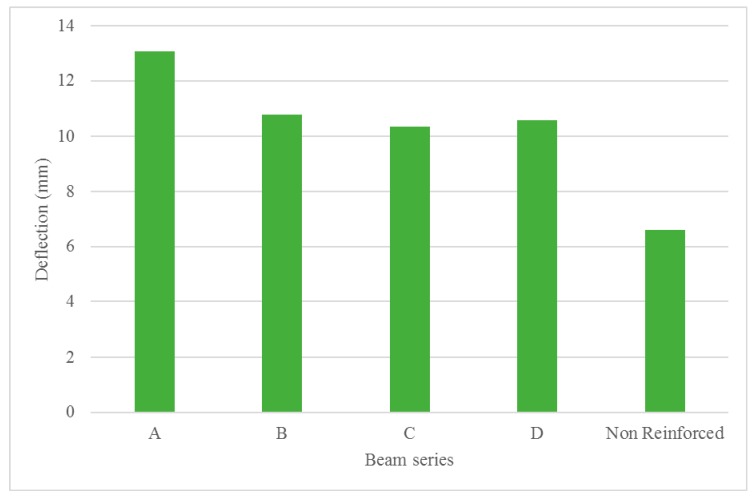
Maximum mid-span deflection of the beams investigated in this study.

**Figure 19 materials-13-00696-f019:**
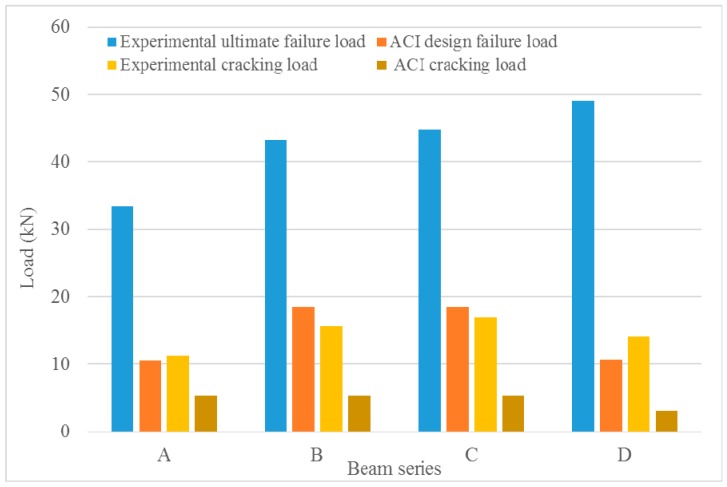
Comparison of the loads obtained through lab tests with the design loads obtained based on ACI 440.1R-15 standard recommendations.

**Figure 20 materials-13-00696-f020:**
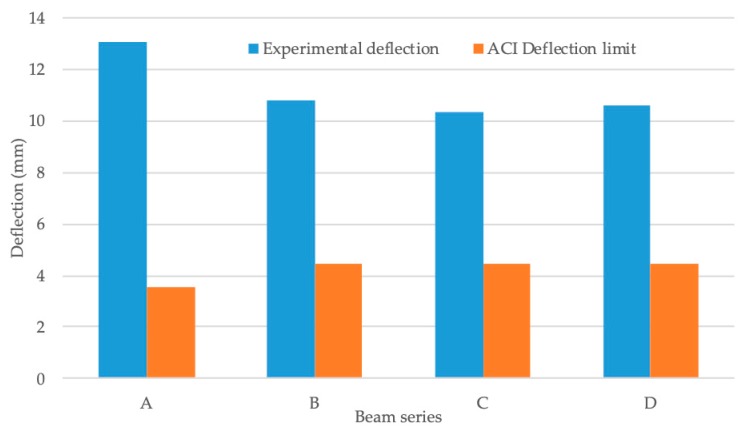
Maximum deflection limits comparison between ACI standard and experiments.

**Table 1 materials-13-00696-t001:** Selected properties of epoxy-resin system investigated in this research.

Property	Value
Tensile Strength	73 MPa
Modulus of elasticity in tension	4067 MPa
Modulus of rupture	125 MPa
Modulus of elasticity in flexure	3624 MPa
Compressive strength	117 MPa
Mix ratio	100 to 33 by weight
Mixed Specific Density	1.105
Pot Life at room temperature	30 min
Min curing time @ 80 °C	20 min
Required total curing time @ 25 °C	7 days
Bio-based content	20% by weight

**Table 2 materials-13-00696-t002:** Concrete mix proportions.

Mix Characteristics	Value	Unit
Water	182	kg/m^3^
Cement	280	kg/m^3^
Sand	850	kg/m^3^
20 mm Aggregates	1050	kg/m^3^
Water/Cement	0.65	
Density	2362	kg/m^3^
Slump	100	mm

**Table 3 materials-13-00696-t003:** Details of the bamboo composite reinforced concrete beams.

Beam Design	Number of Bottom Reinforcement	Number and Spacing of the Stirrups at Each Side of Beam (mm)	Distance between Load Introduction Points (mm)
A	2	6 @ 70	350
B	4	6 @70	350
C	4	4 @ 115	350
D	4	4 @ 75	600

**Table 4 materials-13-00696-t004:** Average properties of bamboo Dendrocalamus asper used in this study [34].

Mean Specific Density	Mean Tensile Strength along the Fiber (MPa)	Mean Tensile Modulus of Elasticity (MPa)	Mean Modulus of Rupture (MPa)
0.72–0.91	216–323	19600–26110	130–205

**Table 5 materials-13-00696-t005:** Average mechanical properties of bamboo composite reinforcement systems produced in this study.

Property	Longitudinal Reinforcement	Transverse Reinforcement (Stirrups)
Tensile Strength (MPa)	346 ± 25	275 ± 26
Modulus of elasticity in tension (MPa)	33,100 ± 3200	30,800 ± 2900
Modulus of rupture (MPa)	262 ± 22	244 ± 20
Modulus of elasticity in flexure (MPa)	29,200 ± 2900	24,900 ± 2200
Compressive strengthalong the fiber (MPa)	162 ± 12	137 ± 11

**Table 6 materials-13-00696-t006:** Bond strength values for bamboo composite reinforcement.

Bond Strength (MPa)	Mode of Failure	Embedment Length (mm)
3.68 ± 0.23	Bamboo composite failure	200

**Table 7 materials-13-00696-t007:** Summary of results obtained for the four-point flexural test of concrete beams.

Beam Design	Specimen	Reinforcement Ratio	Initial Cracking Load	Ultimate Failure Load	Initial Cracking MOR	Ultimate MOR
		(%)	(kN)	(kN)	(MPa)	(MPa)
A	1	0.93	10.2	32.8	2.6	8.4
	2		11.6	32.1	3	8.2
	3		12.1	34.3	3.1	8.8
	4		11.3	34.8	2.9	8.9
B	1	1.86	15.8	44.1	4.1	11.3
	2		16.3	41.3	4.2	10.6
	3		15.1	43.1	3.9	11
	4		15.5	44.8	4	11.5
C	1	1.86	18.9	46.4	4.8	11.9
	2		14.5	40.8	3.7	10.4
	3		17.5	47.8	4.5	12.2
	4		16.8	44.3	4.3	11.3
D	1	1.86	13.5	48.4	2	7.1
	2		15.2	49.3	2.2	7.2
	3		14.8	52.3	2.2	7.7
	4		13.1	46.4	1.9	6.8
Non-reinforced	1	0	4.5	6.3	1.2	1.6
	2		4.9	6.8	1.3	1.7
	3		4.7	6.1	1.2	1.8
	4		4.4	6.5	1.1	1.7

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
