# Peer review of "Application of Sustainable Bamboo-Based Composite Reinforcement in Structural-Concrete Beams: Design and Evaluation"

_materials, 2020, doi:10.3390/ma13030696_

Round 1

Reviewer 1 Report

The paper deals with an up to date topic which fits with the scope of the journal. The idea of the researches and obtained results are interesting. However, there are still some concern must be addressed before it can be accepted for publication. In details my comments are as follows:

1. English language and style are fine/minor spell check required;

2. Abstract: the author should pay more attention to the result and conclusion;

3. What is the novelty of the experimental work?

4. Conclusion: to reiterate the main limitations and constraints from similar researches worldwide.

Author Response

The authors would like to thank the reviewers for their comments. The paper has been  improved as a result. 

Reviewer 2 Report

The paper describes a newly developed bamboo-based composite material for reinforcement in structural-concrete beams. The data obtained are interesting from the point of view of research aimed to find replacement of steel reinforcements. It, however, suggests that the title needs amendment from “bamboo-composite reinforcement” to “bamboo-based composite reinforcement”. Moreover, the generic conclusion on suitability of bamboo-based composite for construction is not fully supported by data obtained and long-term durability has not been even studied. It shall be therefore amended: the statement in the Conclusions chapter on “This research demonstrates the suitability of the newly developed bamboo composite material for use…” needs amendment to “This research demonstrates the potential of the newly developed bamboo composite material for use…”. The last sentence (This results of this study can be utilized for construction of low-cost and low-rise housing units where the demand for ductility is low and where the secondary elements failure provides adequate warning of collapse.) has to be removed as it may mislead to potentially collapsing structures being constructed as fully safe. Paper data could be utilized for developing construction materials rather than constructing houses.

Also, methodologically the paper lacks microstructural and comparison data with similar steel-based reinforcements, such as data of Table 5 on tensile strength that are given without comparison with conventionally reinforced concrete.

There are several minor comments that also need account on revising:

Figure 1: add a scale bar

Table 4: The “Mean Specific density (MPa)” cannot be in MPa, it needs correction to kg/m3.

Author Response

(The authors gave the same response as above.)

Reviewer 3 Report

The article deals with a very actual issue of the replacement of steel reinforcements in concrete beams. Experimentally tested composite reinforcement based on epoxy resin and bamboo fibers is an alternative to steels. Despite some advantages of the composite reinforcement, its use seems to be very limited so far.

I think the article is well written. It contains the results of the specific applied research.

Several parts of the article could be discussed. The properties of the use composite are depending on several factors. These vary widely.

I have several comments and questions to the authors:

At the beginning you write that the strength of bamboo fibers is up to 2000 MPa, the fibers you use have a strength of only 216 to 323 MPa. Wouldn't it be appropriate to use bamboo fibers of higher strength? On page 4 you write that the thicknesses of the fibers was in the range of 0.4 to 1 mm. Were the fibers sorted by thicknesses? It is necessary to specify in the text. How was the homogeneity of the deposited bamboo fibers in the matrix ensured? This needs to be added in the text. How was the composite reinforcement connected to the reinforcement system done? In my opinion, the method and stiffness of the joints have a considerable influence on the resulting properties of the beam. This needs to be added in the text. In Figure 17, I would recommend indicating the relative elongation, not absolute, on the x-axis.

Author Response

(The authors gave the same response as above.)

Round 2

Reviewer 2 Report

Figures 2 and 3 are now wrongly denoted in the manuscript revised as Figures 1 and 2, it is just a misprint to be amended